# Chlorine Gas Sensor with Surface Temperature Control

**DOI:** 10.3390/s22124643

**Published:** 2022-06-20

**Authors:** Andrzej Krajewski, Shadi Houshyar, Lijing Wang, Rajiv Padhye

**Affiliations:** 1School of Fashion and Textiles, RMIT University, Brunswick, VIC 3056, Australia; andrew.krajewski@rmit.edu.au (A.K.); rajiv.padhye@rmit.edu.au (R.P.); 2School of Engineering, STEM College, RMIT University, Melbourne, VIC 3000, Australia; shadi.houshyar@rmit.edu.au; 3Defence Materials Technology Centre, Hawthorn, VIC 3122, Australia

**Keywords:** gas sensor, sensing application, toxic vapor, chlorine sensitive nanomaterial, heater control

## Abstract

The work describes the design, manufacturing, and user interface of a thin-film gas transducer platform that is able to provide real-time detection of toxic vapor. This proof-of-concept system has applications in the field of real-time detection of hazardous gaseous agents that are harmful to the person exposed to the environment. The small-size gas sensor allows for integration with an unmanned aerial vehicle, thus combining high-level mobility with the ability for the real-time detection of hazardous/toxic chemicals or use as a standalone system in industries that deal with harmful gaseous substances. The sensor was designed based on the ability of thin-film metal oxide sensors to detect chlorine gas in real time. Specifically, a concentration of 10 ppm of Cl_2_ was tested.

## 1. Introduction

Chlorine gas has been used for various military, industrial, and domestic applications for a very long time. Exposure to this reactive gas can result in a toxic effect on the respiratory system of humans or animals [1]. Spills in industrial plants can cause major emergencies and evacuation of the affected area [2]. A compact early chlorine gas warning system is desirable in such emergencies.

There have been some publications covering Cl_2_ sensors. The work that has been published so far focuses mainly on the Cl_2_ sensor materials. Joshi et al. [3] discussed polypyrrole (PPy) composite films modified with ZnO nanowires as the sensitive and selective chlorine gas sensor. They achieved high sensitivity at 10 ppm and a fast response time of 55 s.

Baron, Narayanaswamy, and Thorpe [4] mixed silicone rubber with fluorescent porphyrin H_2_TPP to detect Cl_2_ at ppm levels. The produced fluorescent emission effect was low. This sensor is also susceptible to other substances, such as hydrogen chloride, nitrogen dioxide, and nitric acid, making it ineffectual as a selective Cl_2_ gas sensor.

Rao, Godbole, and Bhagwat [5] analyzed a sensor composed of palladium-doped nickel ferrite thin films. Pd doping in a NiFe_2_O_4_ thin film increases the sensitivity to the Cl_2_ gas, thus decreasing the operating temperature of the film. Miyata, Kawaguchi, Ishii, and Minami [6] discussed highly sensitive Cl_2_ gas detection using Cu–phthalocyanine thin films and found that the film conductivity is proportional to the amount of Cl_2_ gas exposure. High sensor sensitivity was achieved when tested at temperatures of around 400 °C. However, such a high working temperature limits its use. Debnath et al. [7] used thin cobalt phthalocyanine films grown on sapphire substrates as a Cl_2_ sensor. The response time of 18 s is short; however, the linear sensitivity response of the composition is only within the range between 0.005 and 2 ppm.

The optical spectrum absorption of thin porphyrin Langmuir–Blodget coated films was elaborated by George et al. [8]. They discovered the absorbance at 460 nm of the light spectrum for the presence of Cl_2_ gas. An optical type of Cl_2_ gas sensor was discussed by Ansari, Karekar, and Aiyer [9]. They concluded that the optical guide cladded with RbCl and AgCl exhibits good selective chlorine-gas-sensing abilities. However, complex equipment is required to detect the changes in the optical signal.

Zhang et al. [10] investigated three-dimensional open porous Tin dioxide (SnO_2_) structures for Cl_2_ sensing properties. They examined a variety of SnO_2_ based structures and achieved up to 5 ppm Cl_2_ at 160 °C. They associated the performance with the small grain size of the sensing structure. A similar Cl_2_ sensitivity performance was reported by Ma et al. [11] and Chu and Cheng [12]. They discussed the performance of Indium-based structures. They achieved sensitivity up to 10 ppm chlorine gas at a low operational temperature (160 °C).

The Indium Oxide (In_2_O_3_)/SnO_2_ heterojunction microstructures as Cl_2_-gas-sensing structures were analyzed by Wang et al. [13]; and Li, Fan, and Cai [14]. They discovered that the In/Sn = 12:1 (molar ratio) sample exhibited a superior response to the chlorine gas and a short recovery time. Furthermore, the p-type semiconductor (Nickel ferrite) was examined by Yang et al. [15] and Gopal Reddy, Manorama and Rao [16]. They discovered that this structure detects the lower concentration of Cl_2_ (10 ppm).

Based on the literature review, the current findings provided a reason for developing a compact and integrated Cl_2_ gas sensor with a heating element and primary user interface. This work introduces a small and low-weight chlorine gas sensing system for possible use in commercial applications. The procedure of designing a lightweight substrate heater for the gas sensing material is described in this paper. The size and simplicity of the target system with a simple self-regulating heating function were chosen for integration with the sensor.

The literature review provided a selection of suitable sensitive materials that can detect Cl_2_ gas in the 1–200 ppm concentration range. The materials, Indium–Tin heterojunction (In-Sn HN), Indium Oxide nanosheets (In_2_O_3_ NS), and Nickel Oxide nanoparticles (NiO NP), with operating temperature ranges between 160 and 260 °C, were chosen for further examination. The Ag-based heater with the Ag-based feedback resistor was used to set and control the sensor’s temperature as the novelty in this work. Rare metals, such as Au, Pt, etc., were used in such applications. This paper focuses on designing and assembling the Cl_2_ gas sensor with a heating element. A resistive feedback component embedded into the ceramic was used to balance the sensor’s working temperature at the required level.

## 2. Sensor Fabrication and Characterization

Each active material requires a heater substrate to perform its sensing operation. Therefore, the green sheets of a Low-Temperature Cofired Ceramic (LTCC) substrate (Ferro Taiwan) were used to manufacture the chlorine gas-sensor prototypes. The fabrication process was completed as follows: shape design, the formation of blank ceramic layers, screen-printing of silver paste, ultraviolet (UV)-laser etching of prototype designs and internal cutout processes, alignment, hard lamination, and, lastly, the external cutout process and firing of ceramic devices. The fabrication process is illustrated in the flowchart in Figure 1.

The main heating element (Figure 2) was designed within the middle of the transducer’s inner section, using a meandering design with 50 µm track/gap features. The substrate heater was designed on a 4 × 4 mm area. The outer section comprises a 500 µm–thick frame for electrode track placement. The rest sections between the frame and inner section were physically cut out for thermal isolation and weight reduction. The design was performed by using CAD software (Figure 2). The main heating element was designed within the middle of the transducer’s inner section, using a meandering design and 50 µm track/gap features.

The prototype ceramic layers were formed from four 80 µm–thick green sheets (Ferro Taiwan). A mesh-screen printing mask with 200 mesh and the desired pattern was used to print a 15 µm–thick film. A blank screen was used to spread silver (Ag) paste (“Ag conductive paste”, Ferro Taiwan) across the ceramic layer. The device pattern was then etched away using a UV laser engraver (15 W 405 nm pulsed laser, DCT, made in Germany). The internal cutouts for the ceramic layer were also made by using the UV laser engraver with the power setting to 60% at 120 kHz.

The ceramic layers were manually aligned for multi-structure devices based on visually matching the multilayer fiducial. Each layer was taped onto an aluminum backing plate with masking tape to affix its position. Next, hard lamination of the multilayer ceramic device was performed with the iso-static laminator across a two-stage process. The first stage of the process is a soft lamination step, using the following settings: 65 °C, 44.82 bar for 5 min. The second stage maintains the temperature (65 °C) but increases the pressure to 68.94 bar for another 5 min. After the hard lamination process, the unfired ceramic devices were cut out using a long ceramic edge cutting machine (Ceramic Cutter CT08003, Pacific Trans Company, Los Angeles, CA, USA). The final step in fabricating the LTCC ceramic device is to “fire” the devices in a high-temperature furnace. The firing process for the LTCC consists of a 6-step heating cycle that is visualized in Figure 3: (i) ramp up to 540 °C at a rate of 2 °C/min, (ii) dwell at 540 °C for 4 h, (iii) ramp up to 900 °C at a rate of 5 °C/min, (iv) dwell at 900 °C for 1 h, (v) ramp down to 800 °C at a rate of 3 °C/min, and then (vi) deactivate heating coils/element and cool naturally within furnace or oven for 14 h.

Using a firing temperature of 900 °C, several devices were fabricated with fully functioning substrate heaters and temperature sense layers (Figure 4). Any successfully fired ceramic device turned a deep green color and shrunk by 18 ± 2% from its original size in line with the manufacturer’s specification. That was taken care of at the CAD design level by increasing the dimensions of the printed layers by 18%.

The next stage is to develop the chlorine gas–sensitive nanomaterials to be placed onto the sensor substrate area (Figure 2). The sensor was then electrically characterized in air and under varied chlorine gas concentration conditions.

## 3. Chlorine Gas–Sensitive Nanomaterial Fabrication Procedure

The gas-sensitive materials In_2_O_3_ NS, In-Sn HN, and NiO NP were synthesized. All ingredients of analytical grade were purchased from Sigma-Aldrich and were used without further refinement.

### 3.1. Nickel Oxide Nanoparticle (NiO NP) Synthesis

The NiO NP synthesis method was observed by Arif, Sanger, and Singh [17]. First, chill 100.0 mL ethanol to 5 °C, and then add 42 mg (1 mol LiOH.H_2_O) lithium hydroxide monohydrate to the chilled ethanol and mix until dissolved. Then slowly add, while stirring, 124.42 mg (0.5 mol nickel acetate tetrahydrate (Ni(CH_3_COO)_2_.4H_2_O) until the material dissolves. Store the resultant mixture at 5 °C for 3 days, during which the NiO NP precipitates out of the solution. The NiO NP was separated via centrifuge (4000 RPM) and washed with deionized (DI) water three times. The precipitate was calcinated at 400 °C for 2 h. The powdered contents were then transferred to 20 mL of DI water and ethanol (1:1 *v*/*v*).

### 3.2. Indium Oxide Nanosheets (In_2_O_3_ NS) Synthesis

The In_2_O_3_ NS synthesis method was observed by Ma et al. [18]. Indium(III) nitrate hydrate (In(NO_3_)_3_·4.5H_2_O), analytical reagent grade (≥99.5%), and commercial melamine-formaldehyde (MF) sponge were purchased from Sigma-Aldrich. Prior to use, the MF sponge was ultrasonically cleaned in absolute ethanol for 30 min and dried at 80 °C.

In(NO_3_)_3_·4.5H_2_O (1 g) was added to 160 mL of distilled water and stirred to obtain a clear solution. Then a small MF sponge (2.4 g) was soaked in this solution for 3 h. Then the MF sponge was brought out and placed in a vacuum oven at 80 °C overnight. Finally, In_2_O_3_ nanosheets were obtained by calcinating the dried MF sponge using a vacuum oven at 650 °C for 3 h at a ramping rate of 3 °C·min^−1^. The powdered contents were then transferred to a 20 mL solution consisting of DI water and ethanol (1:1 *v*/*v*).

### 3.3. In_2_O_3_/SnO_2_ Heterojunction Microstructures

The In_2_O_3_/SnO_2_ synthesis observed in this paper was from Li, Fan, and Cai [14]. In(NO_3_)_3_·4.5H_2_O (0.3819 g, 1.0 mmol) and SnO_2_ (21.53 mg, 0.1428 mmol) were blended in an agate mortar and ground thoroughly for 15 min at room temperature. Then NaOH (0.1600 g, 4.0 mmol) was added to the mixture and ground for 45 min.

Removal any unreacted reagents and by-products was performed via centrifuge and washing twice with DI water and once with absolute ethanol (4500 RPM for 5 min). The resultant was then placed into a glass Periti dish and dried at 80 °C for 12 h. After drying, the mixture was calcined at 500 °C for 2 h. The powdered contents were then transferred to a 20 mL solution of DI water and ethanol (1:1 *v*/*v*).

### 3.4. Transferring the Synthesized Nanomaterials onto the LTCC Sensor Prototypes

The gas-sensitive materials were applied via drop-casting 10 µL of the nanomaterial solution onto the LTCC sensor prototype (on the sensor substrate layer) and allowed to air-dry.

## 4. Fabrication of Temperature Control System

The functioning prototype aims for good thermal performance, high heating speed, and thermal insulation from the controlling circuitry.

### 4.1. Concept of Using Two Printed (Embedded into Ceramic) Resistors Design

In contrast to most ceramic sensor platforms, this novel design uses the sintered Ag silver paste for the heating and temperature-sensing elements [19]. The two-element design involves one resistor laser etched on an additional layer of the ceramic under the sensor as the heating element and another resistor laser etched at the bottom of the assembly as the Positive Temperature Coefficient (PTC) resistor, which provides feedback to the electronic control unit (Figure 5). The resistance changes were used to supply power to the heater efficiently.

The set of tests conducted used a purposefully built rig containing a Fluke 87 five-digit multimeter measurement of resistance (R_fb_) of the temperature sense layer (also referred to as the “feedback resistance”), a Fluke 117 four-digit multimeter for voltage and current measurements, and an HP6235 power supply. Infrared (IR) emission measurement was conducted using a Testo portable InfraRed camera and analyzed by IRsoft by TESTO. Five samples were tested for temperature versus supplied power changes.

### 4.2. Temperature Control System

A simple temperature-control circuit was embedded in a ceramic PTC resistor. Low-resistance measurement techniques were adopted, including (i) the Voltmeter–Ammeter Method, and (ii) Kelvin’s Double-Bridge Method. A temperature control system used the instrumentation amplifier INA155 (Texas Instruments) and a modified resistor bridge. The feedback-controlled heater is shown in Figure 6.

The substrate heater is supplied directly from the “Power In” V_CC_ through the power driver Q1 (FDN304). Initially, the maximum power was provided to the heater to speed up the heating of the surface under the sensor. However, the sensor’s temperature changes influence the feedback resistor (R_fb_) resistance value. That drives the bridge R1, R2, R3, and R4+R5 out of balance. The differential signal is fed into the INA 155, a high-precision Instrumentation Amplifier that produces voltage levels at the output (6) of the INA155 circuit. The output is fed into input (6) of the comparator LM393 (Texas Instruments). The comparator compares the incoming signal level with the reference voltage corresponding to the designated temperature under the Cl_2_ sensor. The output circuit of the comparator switches the heater off if the sensor’s temperature exceeds the set level or switches it on when the temperature is too low. Using positive-feedback resistor input and output of the comparator, the hysteresis is established to avoid the jitter of the switching signal, as it may produce electrical interference with other parts of the circuitry. Once manufactured and tested, the polyurethane lacquer protected the circuit against a corrosive environment.

The Cl_2_ sensor is set up in a bridge configuration. The circuitry uses INA155 to amplify the signal generated by the unbalanced bridge and feed these changes into the analogue-to-digital converter ADS1112 (Texas Instruments). Any microprocessor can process the digital signal with the I2C serial interface.

## 5. Results and Discussion

The power supplied to the heater is consistent, and it does not significantly vary with the resistance of the heater for all valid devices. The Cl_2_ sensing area (4 mm × 4 mm) has unchanged dimensions for the above experiments. The gradient shift in power supplied to the heater shown in Figure 7a (Device 1, Heater 6.1 Ω) may be explained by the changes in the thickness of the ceramic.

Changes in the value of the feedback resistor (R_fb_) are consistent with the sensor’s surface temperature changes (Figure 7b). Therefore, the R_fb_ is well suited as the PTC thermal sensor for controlling the Cl_2_ sensor’s surface temperature. The relationship between power and the feedback resistor is explained by the physical differences in the thickness of the heater and feedback resistor’s tracks caused by the imperfections of the screen-printing process.

The feedback-controlled heater with the connected sensor weighs 7 g and allows for deployment in portable or deployable systems. The sensor’s temperature was set to 240 °C, and the circuit depicted in Figure 6 was tested under these conditions. In addition, the IR emission was measured across the sensor’s area, and the results are depicted in Figure 8.

The temperature shows consistency across the sensing area, with a maximum change of 28 °C between the far edge and the middle part of the sensor. The circuit allows the substrate temperature to be set up to 320 °C. The thermal isolation between the inner sensing area from the external frame was achieved by cutting out the section between the frame and the sensing area.

The prototype samples were loaded into the Linkam stage (Figure 9). The 10 ppm Cl_2_ gas standard and precision multimeter were connected to the testing rig to assess the sensor’s sensitivity and time responsiveness. The heater was set to 260 °C.

As part of the test setup, a data logging unit was connected to the multimeter to record resistance changes associated with the presence of Cl_2_ gas. The data logging section consists of a custom NI LabVIEW data logging script that logs resistance measured by the multimeter every second. The sensor under test (SUT) was evaluated via the following processes:i.Establishing initial conditions for sensor measurement—The SUT was first exposed to compressed dry air (CA) at a flow rate of 100 SCCM until the temperature of the SUT reached 260 °C.ii.Recording the sensor’s baseline (or reference)—At a temperature of 260 °C, the SUT’s baseline resistance was established by flowing CA at 100 SCCM for around 20 min.iii.Measuring the sensor’s change in response to a toxic gas vapor—The CA was stopped, and 100 SCCM of test gas (10 ppm Cl_2_ in N_2_ bal) was applied for 10 min.

This procedure was repeated for In-Sn HN, In_2_O_3_ NS, and NiO NP. The test results are shown in Figure 10, Figure 11 and Figure 12, where the right-pointing arrow indicates the moment from which the 10 ppm of Cl_2_ gas was applied.

The three graphs feature a SUT’s baseline resistance between 0 and 1367 s. That was established by flowing CA at 100 SCCM for more than 20 min. After this, the CA was stopped, and 100 SCCM, the 10 ppm Cl_2_ gas standard, was introduced to the SUT.

The total resistive change between the baseline and steady state was calculated to evaluate the sensor’s sensitivity. If the sensor did not reach a steady state within the 10 min period, the last data point of the set was used to calculate the sensitivity. The response time measurement was observed via the time taken to reach 90% of the SUT’s steady-state response (or the last data point measured if no steady-state output was reached). The results are shown in Table 1.

Only one of three tested gas-sensitive materials performed with reasonably responsiveness and sensitivity to 10 ppm Cl_2_ gas content. The In_2_O_3_ NS achieved stability within 142 s, with a resistance value between 103 and 271 Ω. The circuit with a bridge configuration and analogue-to-digital conversion described in Figure 6 can be used to convert changes in Cl_2_ gas concentration to the digital signal for further processing.

## 6. Conclusions

The work described in this paper provided a guide and procedures to design an integrated toxic vapor transducer with the heating and temperature sensing element and analogue to digital conversion of gas concentration. The aim was to use it in standalone, handheld, or deployable systems. Three Cl_2_ gas–sensitive materials were chosen to demonstrate the proof-of-concept platform. The manufacturing procedures of these materials were described in detail, and the fabrication of the substrate and its heater was provided. The gas-sensitive materials Indium Oxide nanosheets (In_2_O_3_ NS), Indium-Tin heterojunction (In-Sn HN), and Nickel Oxide nanoparticles (NiO NP) were synthesized and tested. The functioning prototype provided good thermal performance and thermal insulation from the controlling circuitry. The temperature control circuit embedded in a ceramic Ag-based PTC resistor was manufactured and tested. It was concluded that the change of the feedback resistance, R_fb_, corresponds with the performance of the PTC temperature sensor. A reliable and accurate heat supply was provided to the substrate for the gas-sensing compounds to work properly. Only the In_2_O_3_ NS performed within a reasonable time to achieve a steady endpoint of fewer than 10 min. The substrate geometry with the temperature-controlled heater circuitry can be used for any device that requires a substrate with self-adjusting heating for active sensing material.

## Figures and Tables

**Figure 1 sensors-22-04643-f001:**
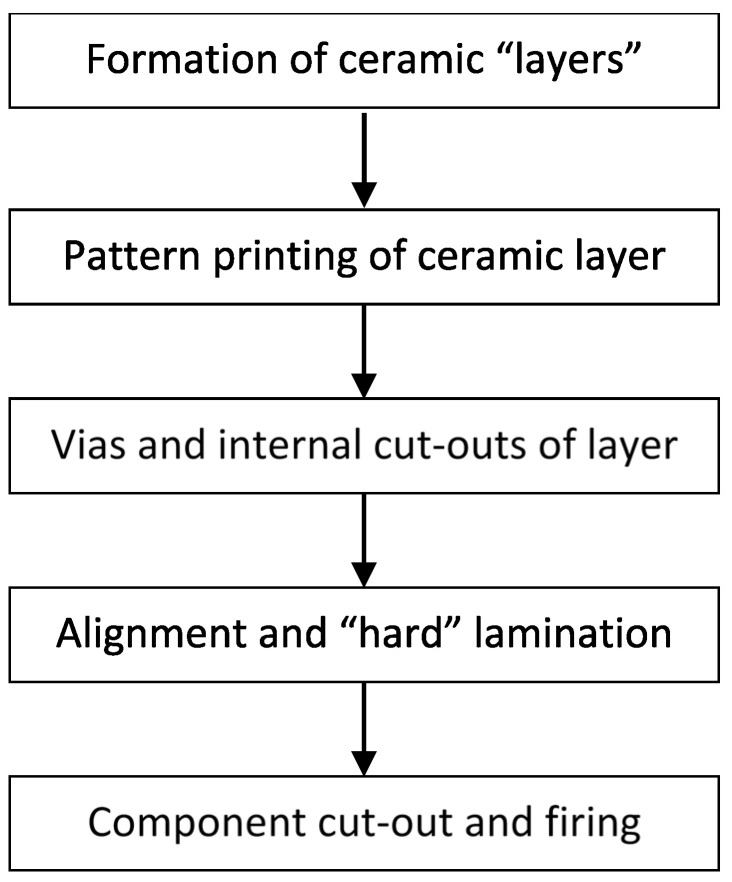
Fabrication process from LTCC ceramic transducer.

**Figure 2 sensors-22-04643-f002:**
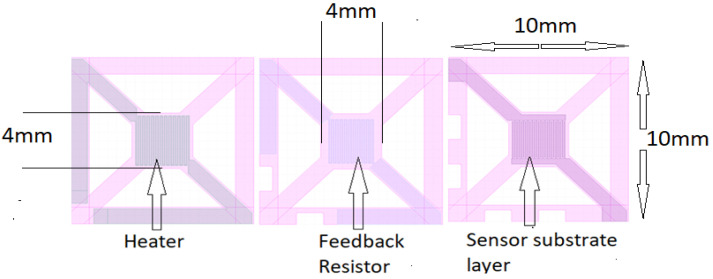
Sensor design.

**Figure 3 sensors-22-04643-f003:**
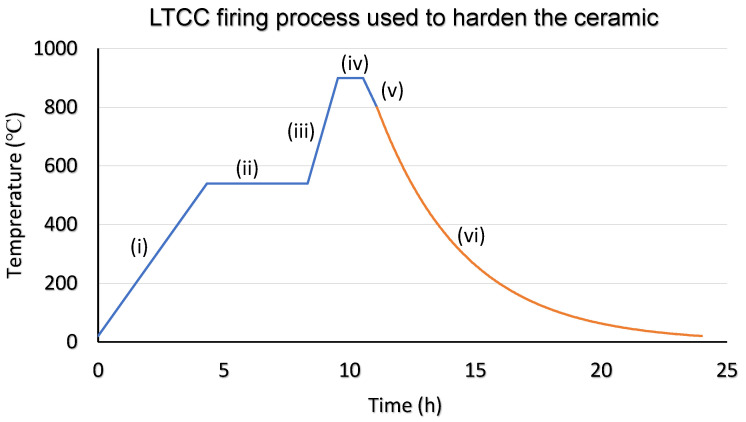
LTCC firing process used in the formation of Cl_2_ gas sensor transducer.

**Figure 4 sensors-22-04643-f004:**
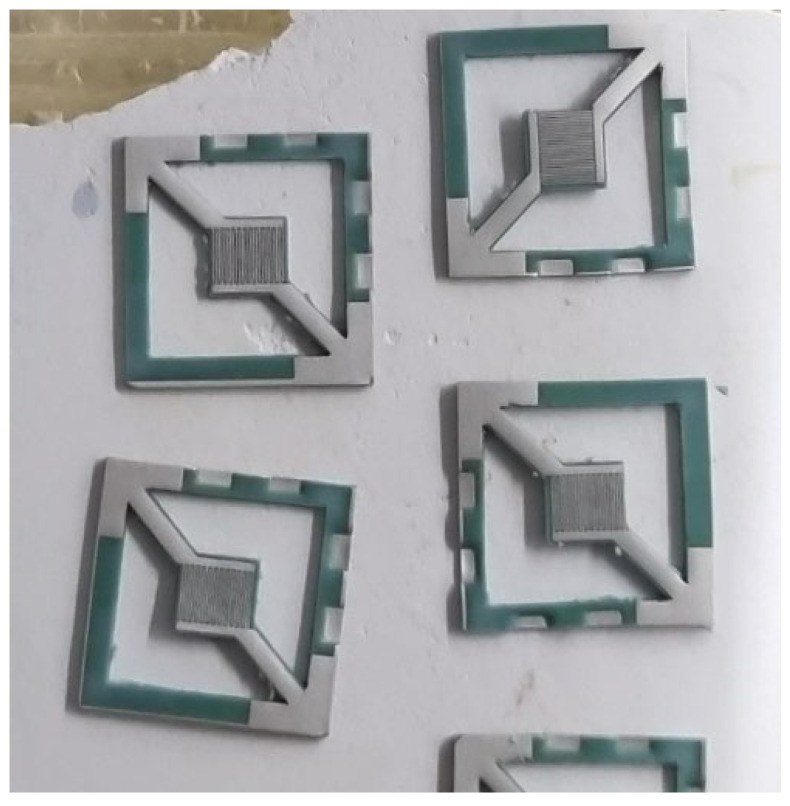
Sensor prototypes after firing.

**Figure 5 sensors-22-04643-f005:**
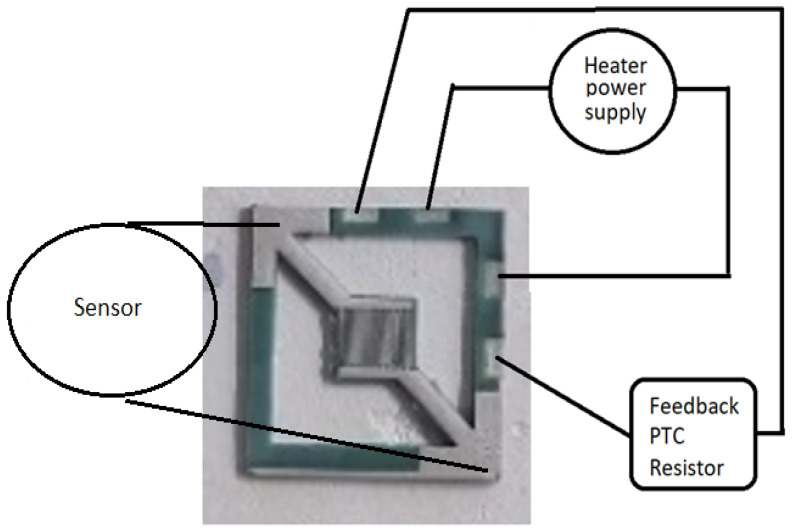
Heater, sensing area, and PTC resistor connections within the ceramic.

**Figure 6 sensors-22-04643-f006:**
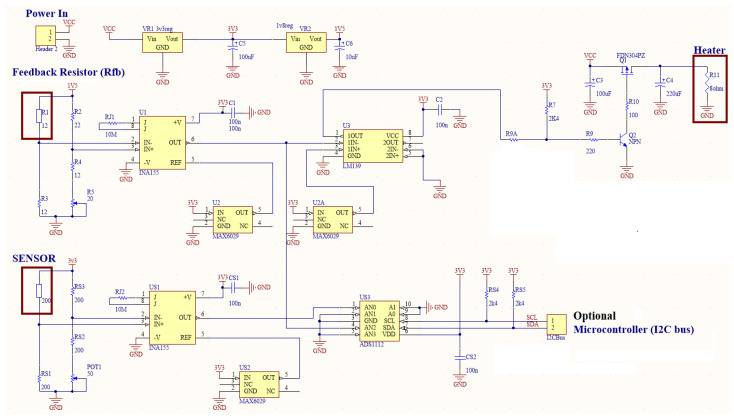
Feedback-controlled heater solution.

**Figure 7 sensors-22-04643-f007:**
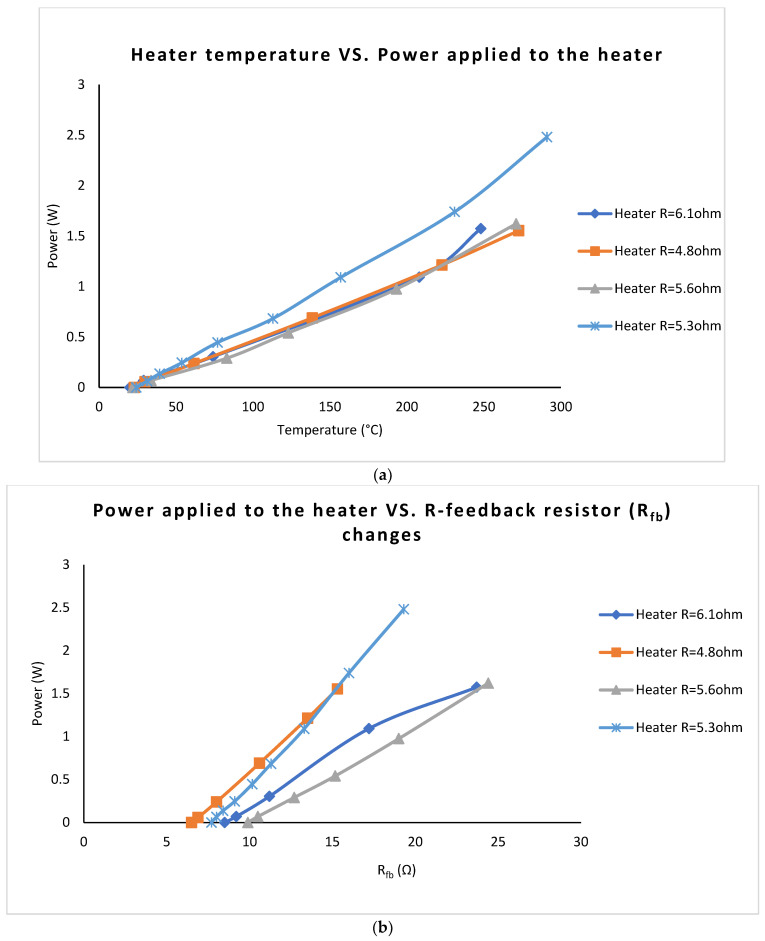
(**a**) Temperature vs. power. (**b**) Feedback resistor’s value variation with power supplied to the heater.

**Figure 8 sensors-22-04643-f008:**
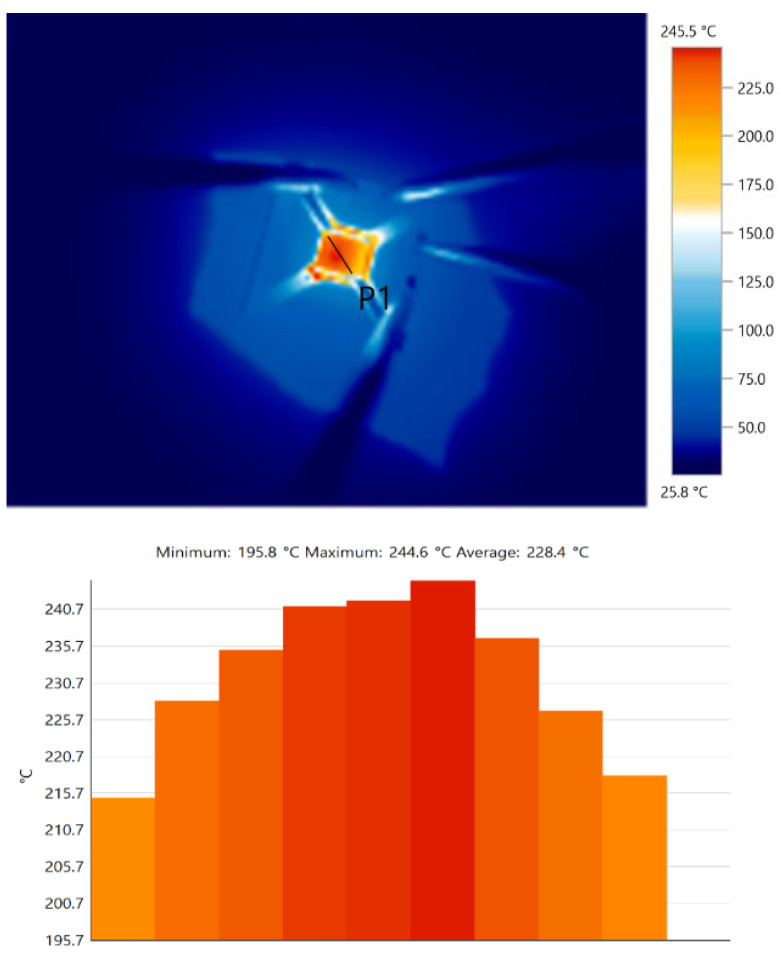
IR image (**top**) and the temperature profile (**bottom**) of the sensor surface with the heater feedback control.

**Figure 9 sensors-22-04643-f009:**
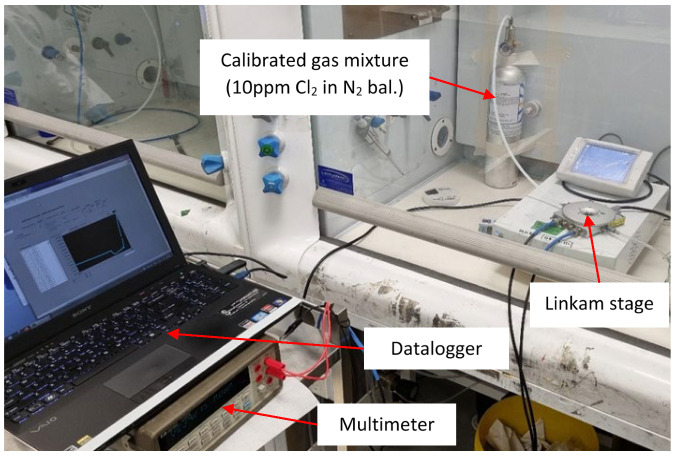
Test setup for the evaluation of Cl_2_ gas–sensitive materials.

**Figure 10 sensors-22-04643-f010:**
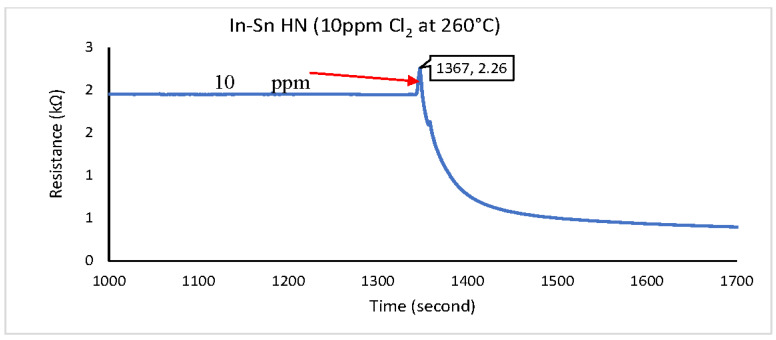
Cl_2_ response for In-Sn HN material at 260 °C.

**Figure 11 sensors-22-04643-f011:**
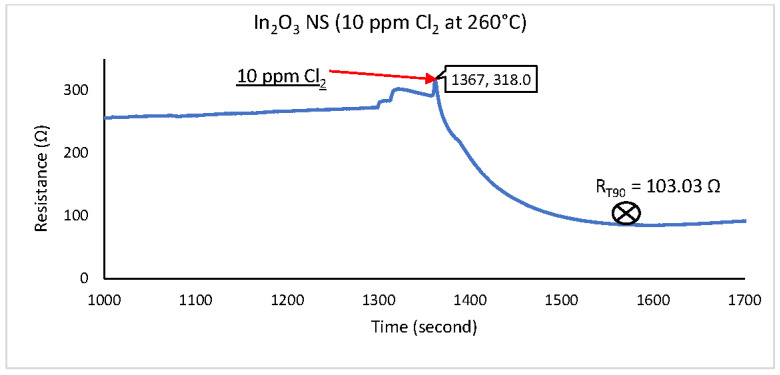
Cl_2_ response for In_2_O_3_ NS material at 260 °C.

**Figure 12 sensors-22-04643-f012:**
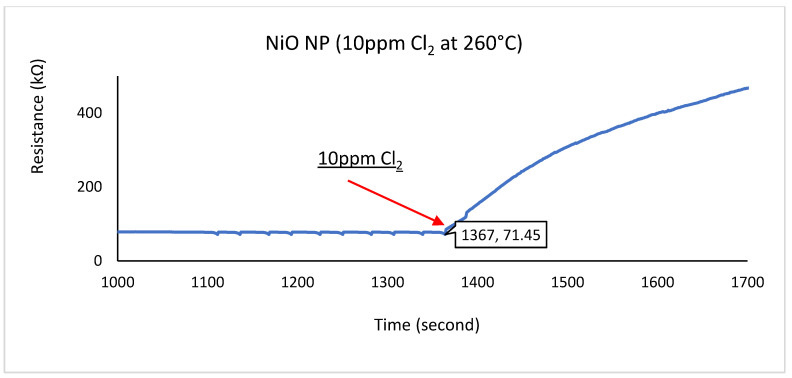
Cl_2_ response for NiO NP the material at 260 °C.

**Table 1 sensors-22-04643-t001:** Summary of Cl_2_ gas sensitive material’s sensitivity and responsiveness to 10 ppm Cl_2_ (in N_2_ bal.) at 260 °C.

Material under Test	Baseline Resistance	Steady State	^A^ Steady-State End Point R $R_{10ppm\;{Cl}_{2}}$	^B^ Sensitivity (ΔR)	Response Time (T90)
IN-SN HN	1.95 kΩ	No	^A^ 396 Ω	79.69%	>10 min
IN2O3 NS	271 Ω	Yes	85 Ω	68.63%	142 s
NIO NP	77.60 kΩ	No	^A^ 466.7 kΩ	501.41%	>10 min

^A^—Endpoint used. Sensor response did not reach steady-state conditions within 10 min. ^B^—$\Delta R=\frac{R_{baseline}\;-\;R_{10ppm\;Cl_{2}}}{R_{baseline}}\cdot 100$%.

## Data Availability

Data may be available upon request.

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
