# Peer review of "Chlorine Gas Sensor with Surface Temperature Control"

_sensors, 2022, doi:10.3390/s22124643_

Round 1

Reviewer 1 Report

In this manuscript, the authors introduced a small and low-weight chlorine gas sensing system for commercial applications. The design and characterization of the sensor are well discussed and demonstrated.  Some minor concerns should be addressed before publication. 

  1. In line 265-267, page 10, there are some words like " errors! reference source not found".
  2. The meaning of the arrow in figure 10 is not clear
  3. Figure 7 a and b should be arranged and marked clearly.
  4. The detection mechanism should be explained clearly.

Author Response

Responses to Reviewer 1

[Sensors] Manuscript ID: sensors-1729521 - Revision

Title: Chlorine Gas-Sensor with Surface Temperature Control

Submitted to: Sensors

Editor, Jameson Zhang; Email: [email protected]

The authors are grateful to the respected editor and honourable reviewers for reviewing our manuscript and highlighting areas that need improvement. Their constructive comments helped us to enhance the quality of the paper. We have revised the paper accordingly and replied to the comments from reviewers as follows:

Reviewer 1

Comments and Suggestions for Authors

In this manuscript, the authors introduced a small and low-weight chlorine gas sensing system for commercial applications. The design and characterization of the sensor are well discussed and demonstrated.  Some minor concerns should be addressed before publication.

In line 265-267, page 10, there are some words like " errors! reference source not found".

Reply: This might be caused by the linked Endnote references when the manuscript is formatted in a difference application. We will remove all reference links to avoid similar errors in the revised version.

The meaning of the arrow in figure 10 is not clear

Reply: We labelled the arrow.

Figure 7 a and b should be arranged and marked clearly.

Reply: We expanded the chart title and formatted the charts.

The detection mechanism should be explained clearly.

Reply: We cited references for the detection mechanism.

In addition to addressing the reviewers’ comments, we also edited the manuscript for English language and style. A track change version is attached to indicate the changes made.

Kindest Regards,

Lijing Wang

Reviewer 2 Report

In this paper, a Cl2 gas sensor with heating element and analog-to-digital conversion of gas concentration is designed and manufactured. Three Cl2 gas sensing materials were were synthesized and tested, that is, indium oxide nanosheets (in2o3-ns), indium tin heterojunction (in SN HN) and nickel oxide nanoparticles (NiO). Only In2O3 ns reaches a steady state within a reasonable time of less than 10 minutes. The substrate with temperature control heater circuit can be used for other sensing applications requring self-adjusting heating.

(1) This paper mainly focuses on the design and assembly of the sensor with heating element. However, the research progress of the sensors with heating elements is not summarized in the Introduction section. Please present a short review in this aspect. In addition, please clarify what is the innovation of this paper.

(2) Three kinds of gas sensing materials are selected in this paper. Please explain why  you select them.

(3) As shown in Figures 7, 10, 11 and 12, the charts were not drawn clearly. Please redraw the figures in the publication-standard of the journal.

(4) In general, the gas sensor needs to be used for a long time. In this paper, there is no repeated data in both the temperature control test and the gas test, so it is impossible to predict the service life of the gas sensor. I suggest the authors should provide at least 3 repeating tests in the figures 10-12.

Author Response

Responses to Reviewer 2

[Sensors] Manuscript ID: sensors-1729521 - Revision

Title: Chlorine Gas-Sensor with Surface Temperature Control

Submitted to: Sensors

Editor, Jameson Zhang; Email: [email protected]

The authors are grateful to the respected editor and honourable reviewers for reviewing our manuscript and highlighting areas that need improvement. Their constructive comments helped us to enhance the quality of the paper. We have revised the paper accordingly and replied to the comments from reviewers as follows:

Reviewer 2

(1) This paper mainly focuses on the design and assembly of the sensor with heating element. However, the research progress of the sensors with heating elements is not summarized in the Introduction section.

Please present a short review in this aspect. In addition, please clarify what is the innovation of this paper.

Reply: We revised the Introduction and highlighted that this paper focuses on the design and assembly of the Cl2 gas sensor with a heating element. A resistive feedback component embedded into the ceramic is used for balancing the sensor's temperature at the required level. The feedback-controlled heater system with the connected sensor weighs 7 g, which is super lightweight and allows deployment in portable or deployable systems.

(2) Three kinds of gas sensing materials are selected in this paper. Please explain why  you select them.

Reply: Based on literature review, we selected (potentially) suitable sensitive materials that can detect Cl2 gas in the 1 – 200 ppm concentration range for evaluation.

(3) As shown in Figures 7, 10, 11 and 12, the charts were not drawn clearly.

Please redraw the figures in the publication-standard of the journal.

Reply: We formatted the figures. High resolution figures/images can be provided for publication.

(4) In general, the gas sensor needs to be used for a long time. In this paper, there is

no repeated data in both the temperature control test and the gas test, so it is impossible to predict the service life of the gas sensor. I suggest the authors should provide at least 3 repeating tests in the figures 10-12.

Reply: This suggestion is very valuable for sensor durability and reliability assurance. However, the sensors were initially designed for single use. Repeating tests could be our future work.

In addition to addressing the reviewers’ comments, we also edited the manuscript for English language and style. A track change version is attached to indicate the changes made.

Kindest Regards,

Lijing Wang

Reviewer 3 Report

Dear Authors,

The article entitled „Chlorine Gas-Sensor with Surface Temperature Control” describes the design, manufacturing of a Cl2 gas sensor. Strong application in a field of Cl2 gas sensing may be applied because of for example small gas sensor sizes. The specialy design support with heater structure may allow for application for other type of gas sensing systems.

The Authors presents the series of electrical, IR imaging, sensor response and stability test. In the present form of manuscript there is a lack of structural analysis of prepared sensors. A lot of nanomaterials fabrication procedures have been applied and there are no results about for example particle sizes, sensor surface properties and structural properties. Those information may be helpful for further sensor design. It could be interesting to have an information from some of proposed methods: SEM, XRD, TEM and XPS.  The set of those data together with data presented in a manuscript will provide sufficient information about designed gas sensors.

The quality of Figures have to be improved:

Figure 4 – improve quality of a photo

Figure 5 – increase font size

Figure 7 a) – the legend is cutted

Figure 8 – increase font size change the color of cross line in IR image (left), the x: scale should be marked with cross section length data

Figure 12 – graphs should be improved, there should be no line crossing, text overlapping.

Page 10 reference error

Why the heater temperature was set to 260C and not 255C or 265C. What was the reason. Is there any explanation of so high difference in baseline resistance between designed gas sensors?

Author Response

Responses to Reviewer 3

[Sensors] Manuscript ID: sensors-1729521 - Revision

Title: Chlorine Gas-Sensor with Surface Temperature Control

Submitted to: Sensors

Editor, Jameson Zhang; Email: [email protected]

The authors are grateful to the respected editor and honourable reviewers for reviewing our manuscript and highlighting areas that need improvement. Their constructive comments helped us to enhance the quality of the paper. We have revised the paper accordingly and replied to the comments from reviewers as follows:

Reviewer 3

A lot of nanomaterials fabrication procedures have been applied and there are no results about for example particle sizes, sensor surface properties and structural properties. Those information may be helpful for further sensor design. It could be interesting to have an information from some of proposed methods: SEM, XRD, TEM and XPS.  The set of those data together with data presented in a manuscript will provide sufficient information about designed gas sensors.

Reply: These comments are very valuable for theoretical investigation, understanding the mechanism and further research and development. The current work reports the design, and fabrication of a standalone, handheld or deployable system, which meets the industry requirement. We have demonstrated the proof-of-concept platform for Cl2 gas detection, where the sensor system works. The feedback-controlled heater system with the connected sensor weighs 7 g, which is super lightweight and allows deployment in portable or deployable systems. We report the chemicals and fabrication process for the sensor system in detail so that the readers can follow the procedures. Further information on other results such as sensor surface properties and structural properties could make the purpose of this manuscript less clear. We believe more information including sensor material characterisation and the discussion of working mechanisms of the sensors should be a separate manuscript for future publication.

The quality of Figures have to be improved:

Figure 4 – improve quality of a photo

Figure 5 – increase font size

Figure 7 a) – the legend is cutted

Figure 8 – increase font size change the color of cross line in IR image (left), the x: scale should be marked with cross section length data

Figure 12 – graphs should be improved, there should be no line crossing, text overlapping.

Page 10 reference error

Reply: We revised/reformatted the figures.

Why the heater temperature was set to 260C and not 255C or 265C. What was the reason. Is there any explanation of so high difference in baseline resistance between designed gas sensors?

Reply: The operating temperature of NiO NP sensors ranges between 160 and 260°C. We used the upper limit, 260°C, as an example to demonstrate that the newly designed sensor system can work normally at the high temperature. Sensor performance evaluation at different temperatures and platform optimisation could be our future work.

In addition to addressing the reviewers’ comments, we also edited the manuscript for English language and style. A track change version is attached to indicate the changes made.

Kindest Regards,

Lijing Wang

Round 2

Reviewer 2 Report

I recommand acceptance of the work in the preset form.

Reviewer 3 Report

  Dear Authors, Thank you for taking into account the proposed comments of the Reviewers.